# An Expectation-Maximization Algorithm for Training Clean Diffusion Models from Corrupted Observations

**Weimin Bai**[1,2,3]      **Yifei Wang**[4]      **Wenzheng Chen**[5,6]      **He Sun**[1,2,3*]

[1] Academy for Advanced Interdisciplinary Studies, Peking University
[2] College of Future Technology, Peking University
[3] National Biomedical Imaging Center, Peking University [4] Yuanpei College, Peking University
[5] Wangxuan Institue of Computer Technology, Peking University
[6] State Key Laboratory of Multimedia Information Processing, Peking University, Beijing, China
{weiminbai, wyf181030}@stu.pku.edu.cn, {wenzhengchen, hesun}@pku.edu.cn

## Abstract

Diffusion models excel in solving imaging inverse problems due to their ability to model complex image priors. However, their reliance on large, clean datasets for training limits their practical use where clean data is scarce. In this paper, we propose EMDiffusion, an expectation-maximization (EM) approach to train diffusion models from corrupted observations. Our method alternates between reconstructing clean images from corrupted data using a known diffusion model (E-step) and refining diffusion model weights based on these reconstructions (M-step). This iterative process leads the learned diffusion model to gradually converge to a local optimum, that is, to approximate the true clean data distribution. We validate our method through extensive experiments on diverse computational imaging tasks, including random inpainting, denoising, and deblurring, achieving new state-of-the-art performance. The code is available at `https://github.com/ai4imaging/EMDiffusoin`.

## 1 Introduction

Diffusion models (DMs) (1; 2; 3) have demonstrated remarkable versatility in capturing complex real-world data distributions, excelling in diverse applications like image generation (4; 5; 6; 7; 8), audio synthesis (9), and molecular design (10). DMs approximate distributions by learning their score functions—the gradient of the log-likelihood of the data distribution $\nabla_{\mathbf{x}} \log p_{data}(\mathbf{x})$. This enables high-quality sample generation by simulating reverse-time stochastic differential equations (SDEs) (2) during inference.

Recently, there has been growing interest in leveraging DMs as priors for computational imaging inverse problems (11; 12; 13; 14; 15; 16), which aim to recover underlying images $\mathbf{x}$ from corrupted observations $\mathbf{y}$. The Bayesian framework for computational imaging defines the posterior distribution of images $\mathbf{x}$ given observations $\mathbf{y}$:

$$p(\mathbf{x} \mid \mathbf{y}) \propto p(\mathbf{y} \mid \mathbf{x})p(\mathbf{x}), \qquad (1)$$

where $p(\mathbf{y} \mid \mathbf{x})$ defines the forward model of observations and $p(\mathbf{x})$ defines an image prior. DMs offer efficient, data-driven priors that outperform traditional handcrafted priors prone to oversimplification and human biases, such as sparsity (17) or total variation (TV) (18; 19).

However, a major limitation of DM-based solvers is their reliance on substantial volumes of high-quality, clean signals for pre-training—a requirement often infeasible in real-world settings, especially

---

[*]Corresponding author

38th Conference on Neural Information Processing Systems (NeurIPS 2024).

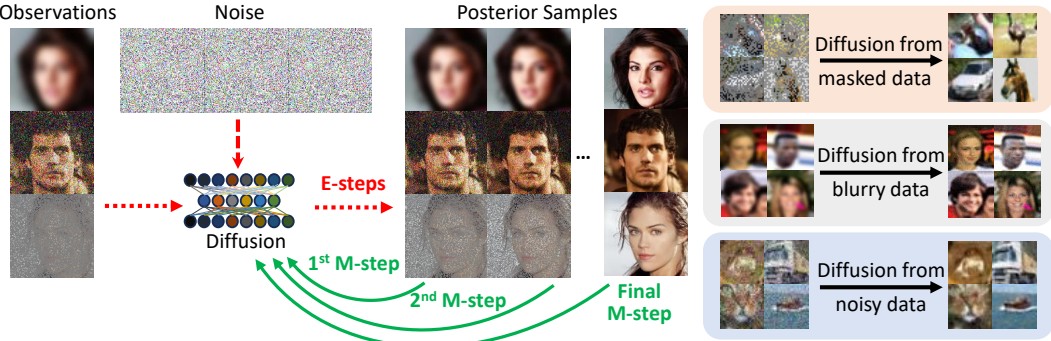

Figure 1: **Overview of EMDiffusion.** The paper proposes an expectation-maximization (EM) approach to jointly solve imaging inverse problems and train a diffusion model from corrupted observations. **Left:** In each E-step, we assume a known diffusion model and perform posterior sampling to reconstruct images from corrupted observations. In the M-step, we update the weights of the diffusion model based on these posterior samples. By iteratively alternating between these two steps, the diffusion model gradually learns the clean image distribution and generates high-quality posterior samples. **Right:** Raw observations and reconstructed clean images based on the diffusion model learned from corrupted data.

for scientific and biomedical imaging. In contrast, corrupted noisy observations with differentiable forward models are easier to acquire, such as blurred images from mobile photography or 2D projections of 3D structures in X-ray computed tomography (CT) (20; 21) and cryogenic electron microscopy (cryo-EM) (22; 23). Our paper seeks to answer a pivotal question: Can a DM be effectively trained to solve inverse problems primarily using large-scale corrupted observations? This presents a chicken-egg dilemma: training an accurate DM requires clean images, but reconstructing clean images from corrupted observations requires a good DM.

Utilizing the Expectation-Maximization (EM) framework, we introduce a novel approach called EMDiffusion. This approach initializes with a diffusion prior trained on a minimal set of clean images, then alternates between two steps across multiple iterations: reconstructing clean images from corrupted observations using the current diffusion prior (E-step), and refining the DM parameters based on these reconstructions (M-step). The sparse clean data provides a good initialization of the DM's manifold, preventing collapse into a distorted or biased distribution characterized solely by corrupted inputs. Each E-M iteration leverages the current diffusion prior to generate cleaner reconstructions from the corrupted data, and these enhanced reconstructions then update the DM, providing an improved prior for the next iteration. This cycle continues, with the generated samples and DM progressively converging toward local optima, which equals to approximate the true clean data distribution. The forward operator and noise process do not affect this type of convergence but only influence the convergence speed by determining the amount of information in the corrupted observations.

We validate the generalizability and effectiveness of EMDiffusion through extensive experiments, applying it to diverse imaging inverse problems across various datasets, including random inpainting, denoising, and deblurring, and achieving compelling results.

## 2 Related Works

**Inverse problems in computational imaging.** Computational imaging aims to reconstruct underlying signals $\mathbf{x} \in \mathbb{R}^d$ from corrupted observations $\mathbf{y} \in \mathbb{R}^m$, where the image formation process is probabilistically modeled as:

$$\mathbf{y} \sim p(\mathbf{y}|\mathbf{x}). \tag{2}$$

Since $m \leq d$ and observation noise is inevitable, inverse problems in computational imaging are ill-posed, with the inverse mapping $\mathbf{y} \rightarrow \mathbf{x}$ being one-to-many. To address this complexity, Bayesian inference introduces a prior distribution of underlying images, $p(\mathbf{x})$, to constrain the solution space for the image posterior, $p(\mathbf{x}|\mathbf{y})$, as illustrated by Eq. 1. Employing Maximum a Posteriori (MAP) estimation, one can derive a point estimate of the underlying image by maximizing $\log p(\mathbf{x}|\mathbf{y})$. Alternatively, posterior image samples of reconstructed images can be obtained through methods like Markov Chain Monte Carlo (MCMC) (24) or Variational Inference (VI) (25; 26; 27). However, the

performance of many computational imaging solvers is limited by their reliance on oversimplified, handcrafted priors such as sparsity and total variation (TV). These priors fail to capture the true complexity of natural image distributions, hindering the solvers' ability to achieve high-quality reconstructions.

**Diffusion models for inverse problems.** Diffusion models (DMs) (1; 2; 3) have recently emerged as powerful data-driven priors for solving imaging inverse problems. By mastering the intricate distribution of images through training on extensive image data, DMs facilitate both point estimates via Plug-and-Play (PnP) optimization (28; 29) and posterior sampling through generative PnP (GPnP) (30), PnP Monte Carlo (PMC) (31), or Diffusion Posterior Sampling (DPS) (13; 14; 32). These approaches have demonstrated remarkable efficacy in addressing a broad spectrum of noisy inverse problems, with applications spanning diverse fields, including astronomy (11; 33) and biomedical imaging (15; 34).

**Learn diffusion models from corrupted data.** In many real-world scenarios, acquiring large-scale clean data is costly or infeasible, motivating efforts to learn DMs directly from corrupted data. Data corruptions stem from under-determined forward models (e.g., 2D projections, inpainting, compressed sensing) and measurement noise. Recent studies have explored various strategies to address these challenges. For instance, in inverse graphics, researchers integrate the forward model into the diffusion process and introduce a view-consistency loss over multiple noiseless projections of the same object to learn a 3D DM from 2D images (35; 36). In image inpainting, AmbientDiffusion (37) randomly masks additional pixels and forces the DM to restore these deliberate corruptions. Since the model cannot distinguish between original and further corruptions, it effectively learns the uncorrupted image distribution. However, the AmbientDiffusion is limited by the additional masking technique and fails to achieve good performance with noisy observations. (38) cleverly finetunes Stable Diffusion (SD) to leverage the pre-trained knowledge in denoising tasks, but does not support training a DM from scratch. Meanwhile, SURE-Score (39) proposes to jointly learn an image denoiser and a score-based DM using Stein's unbiased risk estimate (SURE) loss, where the SURE loss acts as an implicit regularizer on the model weights. Despite its innovative approach, SURE-Score often struggles with significant data corruption, such as inpainting tasks with a large fraction of missing pixels, and tends to produce overly smooth results. A general approach for learning DMs from arbitrarily corrupted data remains an open challenge.

## 3 Preliminary

### 3.1 Score-based Diffusion Models

A diffusion model captures the data distribution by learning a score function, i.e. the gradient of the logarithm of the likelihood of data distribution $\nabla_{\mathbf{x}} \log p_{data}(\mathbf{x})$. Consequently, a diffusion model generates samples by gradually removing noise from a random input, which is equivalent to a reverse-time stochastic differential equation (SDE) - the solution to a forward-time SDE that gradually injects noise,

$$
\begin{aligned}
\text{forward-time SDE:} \quad & \mathrm{d}\mathbf{x}_t = \mathbf{f}\left(\mathbf{x}_t, t\right) \mathrm{d}t + g(t)\mathrm{d}\mathbf{w}, \\
\text{reverse-time SDE:} \quad & \mathrm{d}\mathbf{x}_t = \left[\mathbf{f}\left(\mathbf{x}_t, t\right) - g(t)^2 \nabla_{\mathbf{x}_t} \log p_t\left(\mathbf{x}_t\right)\right] \mathrm{d}t + g(t)\mathrm{d}\overline{\mathbf{w}},
\end{aligned}
\tag{3}
$$

where $t \in [0, T]$, $\mathbf{f}\left(\mathbf{x}_t, t\right) : \mathbb{R}^d \to \mathbb{R}^d$ is the drift function, $g(t)$ controls the rate of the Brownian motion $\mathbf{w} \in \mathbb{R}^d$, and $\overline{\mathbf{w}}$ denotes the Brownian motion running back. A tractable isotropic Gaussian distribution is achieved when $t = T$, i.e. $\mathbf{x}_T \sim \mathcal{N}(\mathbf{0}, \mathbf{I})$, and the data distribution is achieved when $t = 0$, i.e. $\mathbf{x}_0 \sim p_{data}$. $\mathbf{x}_t \in \mathbb{R}^d$ denotes the image $\mathbf{x}_0$ diffused at time $t$. $\nabla_{\mathbf{x}_t} \log p_t\left(\mathbf{x}_t\right)$ is a time-dependent score function, which is usually approximated by a deep neural network, $s_\theta(\cdot)$, parameterized by $\theta$. The generated data distribution from the reverse-time SDE depends only on this time-dependent score function.

### 3.2 Diffusion Posterior Sampling

Many images are consistent with a single observation due to the ill-posed nature of the image formation model. By combining the forward model with the diffusion prior using Bayes' rule, we

define a conditional diffusion process that samples the posterior distribution

$$d\mathbf{x}_t = \left[\mathbf{f}\left(\mathbf{x}_t, t\right) - g(t)^2 \nabla_{\mathbf{x}_t} \log p_t\left(\mathbf{x}_t \mid \mathbf{y}\right)\right] dt + g(t) d\overline{\mathbf{w}}, \tag{4}$$

where the conditional score function can be further decomposed as:

$$\begin{aligned}
\nabla_{\mathbf{x}_t} \log p_t\left(\mathbf{x}_t \mid \mathbf{y}\right) &= \nabla_{\mathbf{x}_t} \log p_t\left(\mathbf{x}_t\right) + \nabla_{\mathbf{x}_t} \log p_t\left(\mathbf{y} \mid \mathbf{x}_t\right) \\
&\simeq \mathbf{s}_{\theta^*}(\mathbf{x}_t, t) + \nabla_{\mathbf{x}_t} \log \int_{\mathbf{x}_0} p(\mathbf{y} \mid \mathbf{x}_0) p(\mathbf{x}_0 \mid \mathbf{x}_t) d\mathbf{x}_0,
\end{aligned} \tag{5}$$

Since the likelihood function is only defined for $t = 0$, the dependence between $\mathbf{y}$ and $\mathbf{x}_t$ is implicit, making $\nabla_{\mathbf{x}_t} \log p_t\left(\mathbf{y} \mid \mathbf{x}_t\right)$ an intractable integral at each diffusion step. Various techniques have been proposed to address this intractable likelihood function, including exactly computing the probability using an ODE flow (11), bounding the probability through an evidence lower bound (ELBO)(33), and approximating the probability using Tweedie's formula(13; 40; 41; 42). To ensure computational efficiency, we adopt the approximation proposed in (13),

$$p_t\left(\mathbf{y} \mid \mathbf{x}_t\right) \simeq p\left(\mathbf{y} \mid \hat{\mathbf{x}}_0(\mathbf{x}_t)\right), \quad \text{where} \quad \hat{\mathbf{x}}_0(\mathbf{x}_t) := \mathbb{E}\left[\mathbf{x}_0 \mid \mathbf{x}_t\right], \tag{6}$$

for diffusion posterior sampling in all the following sections.

### 3.3 Expectation Maximum Algorithm

The Expectation-Maximization (EM) algorithm (43; 44) is an iterative technique for estimating parameters in statistical models involving latent variables. When the true values of the latent variables are unknown, maximum likelihood estimation (MLE) cannot be directly applied to identify the model parameters. Instead, the EM algorithm maximizes a lower bound of the log-likelihood function, derived using Jensen's inequality:

$$\begin{aligned}
\log p_\theta(\mathbf{y}) = \log \int p_\theta(\mathbf{y}, \mathbf{x}) d\mathbf{x} &\geq \int p_\theta(\mathbf{x} \mid \mathbf{y}) \log \frac{p_\theta(\mathbf{y}, \mathbf{x})}{p_\theta(\mathbf{x} \mid \mathbf{y})} d\mathbf{x} \\
&= \int p_\theta(\mathbf{x} \mid \mathbf{y}) \log p_\theta(\mathbf{y}, \mathbf{x}) d\mathbf{x} - \int p_\theta(\mathbf{x} \mid \mathbf{y}) \log p_\theta(\mathbf{x} \mid \mathbf{y}) d\mathbf{x} \\
&= \mathbb{E}_{\mathbf{x} \sim p_\theta(\mathbf{x}|\mathbf{y})}\left[\log p(\mathbf{y} \mid \mathbf{x}) + \log p_\theta(\mathbf{x}) - \log p_\theta\left(\mathbf{x} \mid \mathbf{y}\right)\right] \triangleq \mathcal{L}(\theta),
\end{aligned} \tag{7}$$

where $\mathbf{x}$, $\mathbf{y}$, and $\theta$ denote the latent variables, observations, and model parameters, respectively. The algorithm alternates between two steps:

- **Expectation step (E-step)**: Sample latent variables from the current estimate of the conditional distribution, $\mathbf{x} \sim p_\theta(\mathbf{x} \mid \mathbf{y})$, and compute the expected log-likelihood lower bound $\mathcal{L}(\theta)$.

- **Maximization step (M-step)**: Maximize $\mathcal{L}(\theta) = \mathbb{E}_{\mathbf{x} \sim p_\theta(\mathbf{x}|\mathbf{y})}\left[\log p_\theta(\mathbf{x})\right]$ to update parameters $\theta$.

This iterative procedure allows the EM algorithm to converge to a local maximum of the observed data log-likelihood, making it a powerful technique for estimation problems involving latent variables, such as Gaussian mixture clustering(45), and dynamical system identification(46).

## 4 Proposed Method

Given corrupted observations $\mathbf{y}$ and a known forward model $p(\mathbf{y} \mid \mathbf{x})$, learning DMs from corrupted data is a parameter estimation problem involving latent variables. The latent variables are the underlying clean images $\mathbf{x}$, and the goal is to estimate the DM parameters $\theta$ that govern the image prior $p_\theta(\mathbf{x})$. Consequently, we can leverage an iterative EM approach to reconstruct clean images and train the DM using corrupted data jointly, as described in Fig. 1 and Algorithm 1.

### 4.1 Initialization: Training a Vague Diffusion Model using Limited Clean Images

The Expectation-Maximization (EM) algorithm needs a good initialization to begin its iterative process, as an improper initialization can result in convergence at an incorrect local minimum. While obtaining a large dataset of clean images is difficult, a small set of clean data is often available. This limited clean data can be used to train an initial DM to start the EM iterations. For example, in all

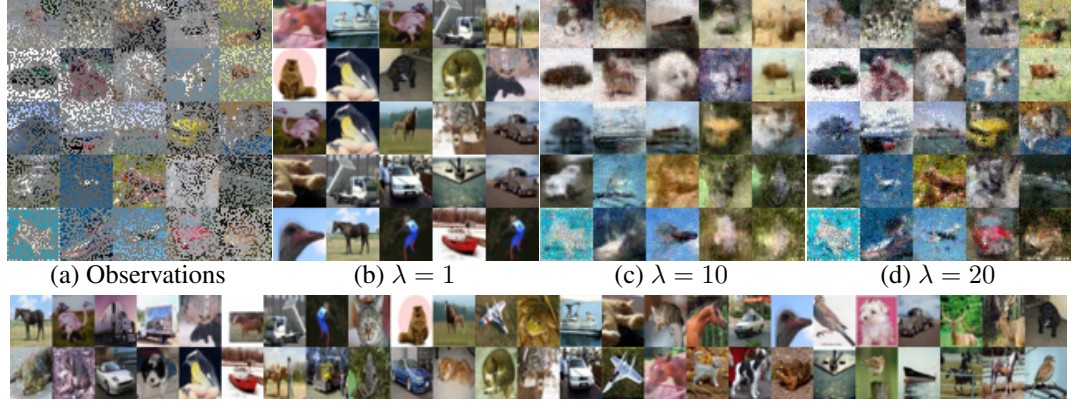

(a) Observations          (b) $\lambda = 1$          (c) $\lambda = 10$          (d) $\lambda = 20$

(e) 50 clean images for training the initial diffusion model

Figure 2: **Adaptive diffusion posterior sampling on CIFAR-10 inpainting.** (a) Corrupted observations from the test set, with 60% of the pixels masked in each image. (b), (c), and (d) Diffusion posterior samples with the diffusion prior weighted by different scaling factors: $\lambda = 1, 10, 20$. The diffusion prior is pre-trained using the 50 clean images shown in (e). When $\lambda$ is small, there is obvious mode collapse, and all posterior samples come from the training set of 50 clean images, unrelated to the observations. As $\lambda$ increases, the data likelihood gains more significance, resulting in reconstructed images that are more consistent with the inpainting observations.

the following experiments, 50 randomly selected clean images were used to train the initial DM, serving as the starting point for the EM algorithm. As demonstrated in Sec. 5.4, clean images do not need to be from the same dataset; those from out-of-distribution datasets also serve as reasonable initializations.

## 4.2 E-step: Adaptive Diffusion Posterior Sampling

In the E-step, we assume a known diffusion prior and reconstruct the underlying clean images through diffusion posterior sampling. We adopt the standard variance-preserving form of the stochastic differential equation (VP-SDE) (2), which is equivalent to the Denoising Diffusion Probabilistic Models (DDPM) (1). The drift function $\mathbf{f}(\mathbf{x}_t, t)$ takes the form $\beta(t)\mathbf{x}_t/2$, and the diffusion rate $g(t)$ is $\sqrt{\beta(t)}$. Therefore, the reverse diffusion sampler in Eq. 4 can be represented as:

$$d\mathbf{x}_t = \left[ -\frac{\beta(t)}{2}\mathbf{x}_t - \beta(t)\nabla_{\mathbf{x}_t} \log p_t\left(\mathbf{x}_t \mid \mathbf{y}\right) \right] dt + \sqrt{\beta(t)}d\overline{\mathbf{w}}, \tag{8}$$

Considering a known imaging forward model, $\mathcal{A}$, and additive Gaussian noise, $p(\mathbf{y} \mid \mathbf{x}) \sim \mathcal{N}(\mathbf{y} \mid \mathcal{A}(\mathbf{x}), \sigma^2 \mathbf{I})$, the conditional score function can be represented as:

$$\begin{aligned} \nabla_{\mathbf{x}_t} \log p\left(\mathbf{x}_t \mid \mathbf{y}\right) &= \nabla_{\mathbf{x}_t} \log p_t\left(\mathbf{x}_t\right) - \nabla_{\mathbf{x}_t} \log p_t\left(\mathbf{y} \mid \mathbf{x}_t\right) \\ &\simeq \mathbf{s}_\theta(\mathbf{x}_t, t) - \frac{1}{2\sigma^2}\nabla_{\mathbf{x}_t} \left\| \mathbf{y} - \mathcal{A}\left(\hat{\mathbf{x}}_0\left(\mathbf{x}_t\right)\right) \right\|_2^2, \end{aligned} \tag{9}$$

where

$$\hat{\mathbf{x}}_0(\mathbf{x}_t) = \frac{1}{\sqrt{\bar{\alpha}(t)}}\left[\mathbf{x}_t + (1 - \bar{\alpha}(t))\,\mathbf{s}_\theta(\mathbf{x}_t, t)\right], \quad \bar{\alpha}(t) = \prod_{s=1}^{t}\left(1 - \beta(s)\right). \tag{10}$$

However, a naive diffusion posterior sampling approach using Eqs. 8, 9, and 10 often fails to produce high-quality reconstructions. This is because the learned DM is inaccurate during the early EM iterations. We demonstrate this issue with a toy experiment. We performed diffusion posterior sampling (DPS) on randomly masked observations, as shown in Fig. 2(a), using an initial DM trained on only 50 clean images. The resulting posterior samples, depicted in Fig. 2(b), show mode collapse due to the severely limited prior. All recovered samples come from the training set of 50 clean images and are unrelated to the observations. Similarly, if the DM is trained on blurry, noisy images with artifacts, naive DPS also performs poorly in image reconstruction.

---

**Algorithm 1** EMDiffusion: Learning Score-based Priors for Inverse Imaging

---

**Require:** DM $\mathbf{s}_\theta$, Observations $(\mathbf{Y}, \mathcal{A})$, few clean data $\mathbf{x}$, Cycles $N$, Timesteps $T$, Epochs $M$,
  Measurement noise $\mathcal{N}(\mathbf{0}, \sigma^2\mathbf{I})$, Diffusion rate $\{\beta_t\}_{t=1}^{T}$
1: Initialize $\mathbf{s}_\theta$ on $\mathbf{x}$ through denoising score matching (47)
2: **for** $i = 1$ to $N$ **do**
3:   $(\mathbf{y}, f) \sim \mathrm{Dataset}(\mathbf{Y}, \mathcal{A})$
4:   $\mathbf{x}_T \sim \mathcal{N}(\mathbf{0}, \mathbf{I})$
5:   **for** $t = T$ to $1$ **do**
6:     $\bar{\alpha}_t = \prod_{s=1}^{t}(1 - \beta_t)$
7:     $\hat{\mathbf{x}}_0 \leftarrow \frac{1}{\sqrt{\bar{\alpha}_t}}\left(\mathbf{x}_t^{(i)} + (1 - \bar{\alpha}_t)\,\mathbf{s}_\theta(\mathbf{x}_t^{(i)}, t)\right)$
8:     $\mathbf{z} \sim \mathcal{N}(\mathbf{0}, \mathbf{I})$
9:     Take reverse-time SDE step on {Sampling in Sec. 4.2}
      $\mathbf{x}_{t-1}^{(i)} \leftarrow \mathbf{x}_t^{(i)} + \beta(t)\left[\frac{\mathbf{x}_t^{(i)}}{2} + \left(\mathbf{s}_\theta(\mathbf{x}_t^{(i)}, t) - \frac{\lambda}{2\sigma^2}\nabla_{\mathbf{x}_t^{(i)}}\|\mathbf{y} - f(\hat{\mathbf{x}}_0)\|_2^2\right)\right] + \sqrt{\beta(t)}\mathbf{z}$
10:   **end for**
11:   **for** $m = 0$ to $M$ **do**
12:     $\mathbf{x}^{data} \sim \mathrm{Shuffle}(\hat{\mathbf{x}}_0^{(i)} \sim p_\theta(\hat{\mathbf{x}}_0^{(i)} \mid \mathbf{y}^{(i)}))$
13:     $t \sim \mathrm{Uniform}(\{1, \ldots, T\})$
14:     $\bar{\alpha}_t = \prod_{s=1}^{t}(1 - \beta_t)$
15:     $\epsilon \sim \mathcal{N}(\mathbf{0}, \mathbf{I})$
16:     Take gradient descent step on {Optimization in Sec. 4.3}
      $\nabla_\theta\left\|\epsilon - \epsilon_\theta\left(\sqrt{\bar{\alpha}_t}\mathbf{x}^{data} + \sqrt{1 - \bar{\alpha}_t}\epsilon, t\right)\right\|_2^2$
17:   **end for**
18: **end for**

---

It does not mean that these low-quality DMs cannot provide any prior information. Although the prior is poor in the early training stages, it has learned common features and structures shared among natural images, such as the continuity and smoothness of natural images and profiles of specific object types. By introducing a hyper-parameter $\lambda$ to rescale the likelihood term and avoid mode collapse, we find that the low-quality DM can also act as a weak prior for posterior sampling, where the reverse-time SDE can be written as:

$$
\begin{aligned}
d\mathbf{x} &= \beta(t)\left[-\frac{\mathbf{x}}{2} - (\nabla_{\mathbf{x}_t}\log p_t(\mathbf{x}_t) + \lambda\nabla_{\mathbf{x}_t}\log p_t(\mathbf{y} \mid \mathbf{x}_t))\right]dt + \sqrt{\beta(t)}d\overline{\mathbf{w}} \\
&\simeq \beta(t)\left[-\frac{\mathbf{x}}{2} - \left(\mathbf{s}_\theta(\mathbf{x}_t, t) - \frac{\lambda}{2\sigma^2}\nabla_{\mathbf{x}_t}\|\mathbf{y} - \mathcal{A}(\hat{\mathbf{x}}_0(\mathbf{x}_t))\|_2^2\right)\right]dt + \sqrt{\beta(t)}d\overline{\mathbf{w}},
\end{aligned}
\tag{11}
$$

The hyper-parameter $\lambda$ efficiently balances the diffusion prior and the data likelihood, resulting in reliable reconstructed images even when the prior is poor. As demonstrated in Fig. 2 (b), (c), and (d), as $\lambda$ increases from 1 to 20, the data likelihood term gains more emphasis, making the reconstructed images more consistent with the inpainting observations. The choice of the hyper-parameter $\lambda$ is automated in each E-step by finding the value that minimizes the data loss,

$$
\lambda^* = \arg\min_\lambda \mathbb{E}_{\mathbf{y}, \hat{\mathbf{x}}_{0,\lambda}}\left[\|\mathbf{y} - \mathcal{A}(\hat{\mathbf{x}}_{0,\lambda})\|_2^2\right],
\tag{12}
$$

where $\hat{\mathbf{x}}_{0,\lambda}$ represents the diffusion posterior samples of reconstructed images with $\lambda$ scaling.

### 4.3 M-step: Optimizing Score-Based Priors

During the M-step, we update the weights of the score-based models using the posterior samples obtained in the E-step. This resembles training a standard clean DM, $\mathbf{s}_\theta$, to approximate the time-dependent score function, $\nabla_{\mathbf{x}_t}\log p(\mathbf{x}_t \mid \hat{\mathbf{x}}_0)$, through denoising score matching (47):

$$
\theta^* = \arg\min_\theta \mathbb{E}_{t, \mathbf{x}_t, \hat{\mathbf{x}}_0}\left[\|\mathbf{s}_\theta(\mathbf{x}_t, t) - \nabla_{\mathbf{x}_t}\log p(\mathbf{x}_t \mid \hat{\mathbf{x}}_0)\|_2^2\right],
\tag{13}
$$

where $t \sim \mathrm{Uniform}(\{1, ..., T\})$, $\hat{\mathbf{x}}_0 = \hat{\mathbf{x}}_{0,\lambda^*}$ represents the posterior samples from the previous E-step, and $\mathbf{x}_t \sim p(\mathbf{x}_t \mid \hat{\mathbf{x}}_0)$ are generated by the forward-time SDE in Eq. 3.

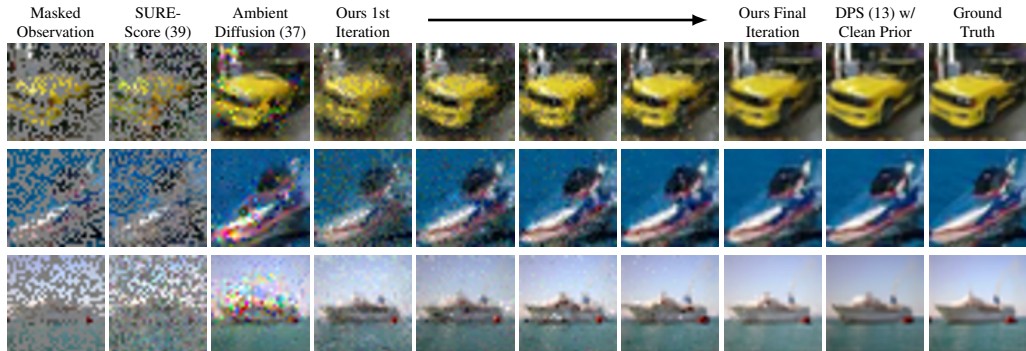

| Masked Observation | SURE-Score (39) | Ambient Diffusion (37) | Ours 1st Iteration | ⟶ | Ours Final Iteration | DPS (13) w/ Clean Prior | Ground Truth |

Figure 3: **Results on CIFAR-10 inpainting.** In each image, 60% of the pixels are masked. As the EM iterations progress, the diffusion model learns cleaner prior distributions, improving the quality of posterior samples. Our method significantly outperforms the baselines, SURE-Score and AmbientDiffusion, achieving reconstruction quality comparable to DPS with a clean prior.

To accelerate training, especially during the early stages when the posterior samples are noisy, the M-step does not always train the score function $s_\theta(\cdot)$ from scratch. In the initial M-steps, we inherit the DM weights from the previous iteration and fine-tune them only using posterior samples from a subset of observations (e.g., randomly select 10% of total observations). However, once the quality of reconstructed images improves sufficiently, we reinitialize the DM weights and retrain the model with 100% data for a few more iterations. The training strategy transitions when the optimal balancing parameter, $\lambda^*$, falls below 1, or fails to decrease for more than three consecutive iterations.

## 5    Experiments

In this section, we demonstrate the performance of our method in learning DMs from corrupted data and solving inverse problems using these models. We validate the method on three imaging tasks: random inpainting, denoising, and deblurring. Our main results are presented in Fig.3, Fig.4, and Table 1, with additional ablation studies in Fig. 5. Further details on neural network architectures, training settings, and additional reconstruction and generation samples are provided in the appendix.

### 5.1    Datasets and Evaluation Metrics

The experiments are conducted on the CIFAR-10 (48) and CelebA (49) datasets at resolutions of $32 \times 32$ and $64 \times 64$, respectively. CIFAR-10 consists of 50,000 images across 10 classes for training, while CelebA contains 30,000 images of human faces. At each iteration, 5,000 corrupted images are randomly chosen for posterior sampling and training, and 250 corrupted images from the test set are chosen for evaluation.

We evaluate the performance of our method using two groups of metrics. First, we compute the peak signal-to-noise-ratio (PSNR) and learned Perceptual Image Patch Similarity (LPIPS) scores between the reconstructed and ground-truth images, quantifying the accuracy of inverse imaging using learned DMs. Additionally, we compute the Fréchet Inception Distance (FID) between the learned DMs and reserved test data to assess their image generation quality.

### 5.2    Baseline and Training Settings

We compare our method with three related baselines: AmbientDiffusion (37), SURE-Score (39), and DPS with clean prior (13). AmbientDiffusion and SURE-Score have similar settings to our method, which do not require DMs pre-trained on large-scale clean signals. Considering AmbientDiffusion is well-designed for masked observations, we only use it as the baseline of the image inpainting task. On the other hand, DPS leverages a pre-trained clean diffusion prior for posterior sampling, so it defines the performance upper bound for our method.

In our experiments, we randomly select 50 clean images from each dataset to train the initial DMs for the EM iterations. AmbientDiffusion is trained with the standard setting in (37). The key hyper-parameter of SURE-Score, $\sigma_\omega$, is set to the observation noise's standard deviation (0.2 for denoising,

Table 1: Numerical Results of inverse imaging and learned priors. The average values of PSNR/LPIPS are from 250 samples randomly selected from the test set. FID is used to evaluate the quality of learned priors by comparing 50,000 generated samples to the train set. Optimal results are highlighted in **bold** and suboptimal results in underline. Note that we take DPS w/ clean prior as the upper bound.

| Method | CIFAR10-Inpainting | | | CIFAR10-Denoising | | | CelebA-Deblurring | | |
|---|---|---|---|---|---|---|---|---|---|
| | PSNR↑ | LPIPS↓ | FID↓ | PSNR↑ | LPIPS↓ | FID↓ | PSNR↑ | LPIPS↓ | FID↓ |
| Observations | 13.49 | 0.295 | 234.47 | 18.05 | 0.047 | 132.59 | 22.47 | 0.365 | 72.83 |
| DPS w/ clean prior | 25.44 | 0.008 | 7.08 | 25.91 | 0.010 | 7.08 | 29.05 | 0.013 | 10.24 |
| Noise2Self (50) | - | - | - | 21.32 | 0.227 | 92.06 | - | - | - |
| SURE-Score (39) | 15.75 | 0.182 | 220.01 | 22.42 | 0.138 | 132.61 | 22.07 | 0.383 | 191.96 |
| AmbientDiffusion (37) | 20.57 | 0.027 | 28.88 | - | - | - | - | - | - |
| Ours | **24.70** | **0.009** | **21.08** | **23.16** | **0.022** | **86.47** | **23.74** | **0.103** | **91.89** |

(a) CIFAR10, Denoising       (b) CelebA, Deblurring

Figure 4: **Results on (a) CIFAR-10 denoising and (b) CelebA deblurring.** Our method significantly outperforms the baseline, SURE-Score, and approximates DPS with clean prior.

and 0.01 for inpainting and deblurring). To ensure a fair comparison, we also provide the same 50 clean images for training AmbientDiffusion and SURE-Score. Details are in Appendix A.

## 5.3 Results

**Image inpainting.** We conduct random inpainting (with mask probability $p = 0.6$) on CIFAR-10. As shown in Fig. 3 and Table 1, our method significantly outperforms AmbientDiffusion and SURE-Score, achieving reconstruction quality similar to DPS with a prior trained on the clean CIFAR-10 dataset. The iterative training process is also illustrated in Fig. 3. Initially, our method performs poorly with the DM trained on only 50 randomly selected clean samples. However, as the E-step and M-step alternate iteratively, the quality of posterior sampling improves. Large-scale posterior samples enrich the priors, leading to enhanced performance at each stage.

**Image denoising.** We perform image denoising on CIFAR-10 with Gaussian noise $\mathbf{n} \sim \mathcal{N}(0, \sigma^2 \mathbf{I})$ and $\sigma = 0.2$. The results are shown in Fig. 4(a) and Table 1. Our method outperforms SURE-Score, and the self-supervised denoising benchmark, Noise2Self (50), though it slightly lags behind DPS with clean priors. However, while our method's reconstructions may appear noisier than DPS results, they sometimes reproduce more details, such as the car wheels in the second row and the cat face in the third row of Fig. 4(a), showcasing the better diversity of our learned DMs.

**Image deblurring.** We validate image deblurring on CelebA using a Gaussian blur kernel with a size of $9 \times 9$ and a standard deviation of $\sigma = 2$ pixels. The results are shown in Fig. 4(b) and Table 1. As with the other tasks, our method significantly outperforms SURE-Score in solving imaging inverse problems, recovering fine details of human faces. However, the FID score of our learned diffusion models lags behind the original blurred observations. This is primarily because the FID score measures image similarity mainly through smooth features, making it a less effective metric for deblurring tasks.

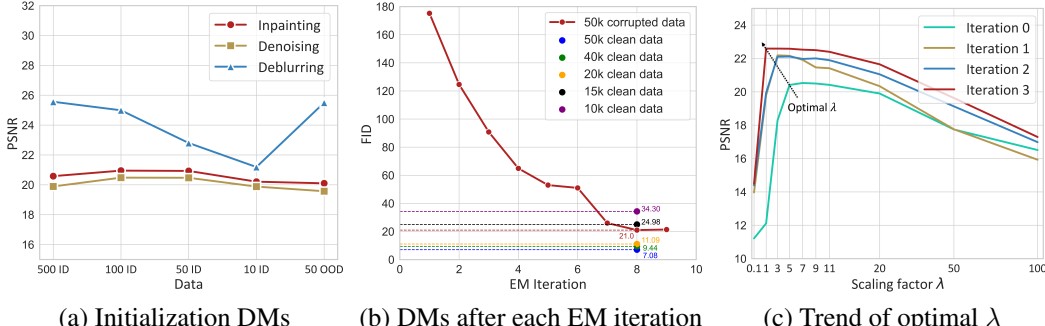

(a) Initialization DMs      (b) DMs after each EM iteration      (c) Trend of optimal $\lambda$

Figure 5: **Ablation studies.** (a) PSNR of diffusion posterior samples generated by the initial diffusion models trained on different amounts (10, 50, 100, 500) or types (in-distribution or out-of-distribution) of clean data. (b) FID scores of learned diffusion models after each EM iteration. The diffusion model trained on 50,000 corrupted images achieves a similar performance to those trained on 15,000-20,000 clean images. (c) PSNR of diffusion posterior samples weighted by different scaling factors $\lambda$ at each stage. The optimal $\lambda$ for posterior sampling decreases as the EM iterations progress.

Comparing the results of all three tasks, we find that AmbientDiffusion only works well for inpainting because its additional masking technique is specifically designed for that purpose. SURE-Score consistently produces over-smoothed results because the SURE loss regularizes the gradient of generated images. As a comparison, our method does not make any special assumptions and provides a general framework applicable to all three tasks. The generation results are in Appendix D.

## 5.4    More Analysis and Ablation Studies

**Number of clean images for training initial DMs.**    Our EM approach starts with DMs trained on a small set of clean images. Fig. 5(a) shows the PSNR of posterior samples generated by these models in the first E-step, allowing us to evaluate the impact of the number of clean training images on the performance of the initial DMs. Remarkably, DMs trained on as few as 10 clean images (0.02% of the corrupted images) can still act as reasonable priors. For inpainting and denoising tasks, DMs trained on 10 clean images provide nearly the same reconstruction quality as those trained on 500 clean images, as these tasks primarily require priors for low-frequency features, and 10 images suffice for an initial guess. However, the deblurring task benefits from DMs trained on more clean data since deblurring aims to recover high-frequency details where more data helps.

Surprisingly, we find that DMs trained on clean images from CelebA (downsampled to 32×32) can also be used to initialize tasks on CIFAR-10. For inpainting and denoising tasks, DMs initialized with 50 clean images from an out-of-distribution (OOD) dataset (CelebA) achieve similar performance to those initialized with in-distribution (ID) data (CIFAR-10). For the deblurring task, DMs trained on OOD data perform better than those trained on a similar amount of ID data, suggesting that OOD data can sometimes serve as stronger priors for guessing high-frequency information.

**Learned priors through iterative training.**    Fig. 5(b) shows the FID scores of the learned DMs in the inpainting task with 50,000 corrupted CIFAR-10 images after each EM stage. The generation ability of the DMs gradually improves as the EM iterations progress. As explained in Sec. 4.3, initially the DM inherits weights from previous steps for fast training and converges at the sixth iteration. After resetting the DM, the training resumes for three more rounds and finally converges to a FID score of approximately 21.08, significantly better than AmbientDiffusion's 28.88, setting a new state-of-the-art. Notably, our method achieves this performance using a DM architecture with far fewer parameters: our method employs a vanilla DDPM with 35.7 million parameters, while AmbientDiffusion uses an improved DDPM++ architecture with over 100 million parameters. Additionally, we compared our method with DMs trained on different amounts of clean data. Our model, learned from 50,000 corrupted images (60% pixels masked) using EM, performs better than a DM trained on around 15,000 clean images.

**Scaling factor in adaptive diffusion posterior sampling.**    Fig. 5(c) shows the PSNR of reconstructed images at each EM stage with different scaling factors, using the inpainting task on CIFAR-10 as an example. We observe that the quality of posterior sampling initially improves and then de-

teriorates as the scaling factor increases, confirming the existence of an optimal scaling factor as suggested in Eq. 12. As the EM stages progress, the optimal scaling factor decreases, indicating that the learned priors progressively improve through the EM iterations. This observation justifies the need for adaptive scaling factors in our method.

# 6 Conclusion

In this paper, we proposed EMDiffusion, a novel expectation-maximization (EM) framework for training diffusion models primarily from corrupted observations. The key assumption is that it is information-theoretically possible to learn the underlying distribution from measurements. Our method demonstrated state-of-the-art performance in image inpainting, denoising, and deblurring across various datasets. Additionally, an important finding is that a small amount of clean, in-distribution data can act as an implicit regularizer, aiding the training of diffusion models from corrupted observations. Future work will aim to 1) extend initialization approaches, potentially by incorporating foundation models or traditional machine learning techniques, such as using preprocessed images from unsupervised inpainting, deblurring, or denoising for initialization, and 2) extend to various imaging inverse problems and learning unknown forward models or noise statistics.

## Acknowledgement

This work was supported by the National Natural Science Foundation of China(62371007) and the National Key Research and Development Program of China (2022YFC3401100).

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

We provide the implementation details and more results in the appendix. We first describe our network architecture and training settings in Sec. A, then show initialization details in Sec. B. More results are also provided in Sec. C and Sec. D.

## A    Implementation Details.

Our neural network architecture follows the vanilla denoising diffusion probabilistic model (DDPM) (1). For quick implementation, see `https://huggingface.co/google/ddpm-cifar10-32` and `https://huggingface.co/google/ddpm-celebahq-256`.

**Model architecture.** Our architecture is exactly aligned with DDPM (1), which is a U-Net (51) based on a Wide ResNet (52). Diffusion time $t$ is implemented by adding the Transformer sinusoidal position embedding into each residual block. For CIFAR-10, our $32 \times 32$ models use four feature map resolutions ($32 \times 32$ to $4 \times 4$) and convolutional residual blocks per resolution level. For CelebA, we increase the feature map number for our $64 \times 64$ to six. We enable the dropout regularization to reduce overfitting. Our CIFAR-10 model has 35.7 million parameters and our CelebA model has 114 million parameters.

**Noise schedule.** We leverage the default settings on VP-SDE (2), which uses a linear schedule with timesteps $T = 1000$, $\beta_1 = 1e - 4$, and $\beta_T = 0.02$.

**Exponential moving average.** To stabilize the training process and reduce the color shift of samples generated by trained DMs, we adopt an exponential moving average (EMA) technique with a decay factor of 0.999 for all experiments.

**Optimizer.** We apply AdamW (53) and set the learning rate to $2e - 4$ for CIFAR-10 and $2e - 5$ for CelebA.

**Hyperparameters for the training process.** We set the batch size to 512 for CIFAR-10 and 64 for CelebA. We set the dropout rate to 0.1 for CIFAR-10 and 0 for CelebA. As for the learning rate, we adapt $1e - 4$ for CIFAR-10 and $2e - 5$ for CelebA, for a larger learning rate will result in unstable training.

**Dataset argumentation.** We only use random horizontal flips to CIFAR-10 during training to achieve better performance. We did not flip CelebA, as the distribution of human faces is quite simple. We provide a table of training hyperparameters in Table 2.

Table 2: Training hyperparameters.

| Dataset | Batch Size | Epoches | LR | Dropout | Optimizer | Data Augmentation |
|---------|-----------|---------|------|---------|-----------|-------------------|
| CIFAR-10 | 512 | 1000 | 2e-4 | 0.1 | AdamW | Random horizontal flips |
| CelebA | 128 | 1500 | 2e-5 | 0 | AdamW | / |

**Baselines.** As for Ambient Diffusion, We use the official checkpoint, which adopts the improved DDPM++ architecture (54) and the EDM scheduler design (55). It also modifies the model architecture's convolutions to Gated Convolutions (56), as they are known to perform better for inpainting-type problems. Their $32 \times 32$ model has 113 million parameters, which is larger than ours. SureScore, on the other hand, uses the deepest NCSNv2 (57) model architecture, which has 95 million parameters. As for DPS with clean priors, we adopt the pre-trained DDPMs provided by `https://huggingface.co/google`. The details of the architectures of all methods are shown in Table 3.

**Training schedule.** At each EM iteration, we randomly choose 5,000 corrupted observations for diffusion posterior sampling and then train DMs. We further divide the iterations into two phases:

• Phase 1 - Resume training DMs: at the early EM iterations, we inherit weights of DMs from the last iteration for quick convergence. For CIFAR-10, this phase lasts for about 6-8 EM iterations, while for CelebA deblurring, this phase increases to 10 EM iterations.

Table 3: Method architecture comparison.

| Model | CIFAR-10 | | | | CelebA | | | |
|---|---|---|---|---|---|---|---|---|
| | Param | Arch | Memory | Schedule | Param | Arch | Memory | Schudule |
| Ours | 35.7M | DDPM | 140MB | VP | 114M | DDPM | 454MB | VP |
| Ambient Diffusion | 113M | DDPM++ | 408MB | EDM | 445MB | DDPM++ | 451MB | EDM |
| SureScore | 90M | NCSNv2 | 1.5GB | VE | 90M | NCSNv2 | 1.5GB | VE |

- Phase 2 - Reset training DMs: at the later EM iterations, we reset the weights of DMs at each M-step, that is, training DMs from scratch. The key insight is that DMs from Phase 1 always have a memory of bad posterior samples, which has a negative effect on the learned distribution. For CIFAR-10, this phase lasts for 3 EM iterations, we found it significantly improves the FID score of DMs. While for CelebA deblurring, this phase lasts for 2 EM iterations until we find the improvement is not obvious.

# B  Additional Strategies for Training Initial DMs

To verify the sensitivity of EMDiffusion's initial DM training data, we provide more quantitative and qualitative results in this section, as shown in Table 4. We draw the conclusion that EMDiffusion is not sensitive to the initial data. Apart from evaluating different numbers of in-distribution (ID) images and out-of-distribution (OOD) images for training the initial DMs, we also test the initialization on preprocessed observations, and find all of them converge similarly.

Table 4: Numerical results of different data for training initial DMs. We show the PSNR values of posterior sampling with diffusion initialized with different data. The results show that EMDiffusion is insensitive to initializations.

| Initialization | Observations | 500 ID | 100 ID | 50 ID | 10 ID | 50 OOD | 50 preprocessed |
|---|---|---|---|---|---|---|---|
| CIFAR10-Inpainting | 13.49 | 20.58 | 20.95 | 20.93 | 20.21 | 20.10 | 16.16 |
| CIFAR10-Denoising | 18.05 | 19.89 | 20.48 | 20.47 | 19.88 | 19.57 | 19.96 |
| CelebA-Deblurring | 22.47 | 25.56 | 24.99 | 22.80 | 21.19 | 25.50 | 22.00 |

Specifically, we preprocess the noisy observations with BM3D (58), the blurry observations with Fast TV Constraint (59), and leave the masked observations unchanged. DMs initialized on these preprocess samples also perform well on the following E-step.

# C  Additional Results on Random Inpainting

Table 5: Comparison of FID scores between the EM approach and Ambient Diffusion across different corruption levels (masking ratio $p = 0.4, 0.6, 0.8, 0.9$) on CIFAR-10 and CelebA.

| Model | CIFAR-10 | | | CelebA | | |
|---|---|---|---|---|---|---|
| | 0.4 | 0.6 | 0.8 | 0.6 | 0.8 | 0.9 |
| Ambient Diffusion | 18.85 | 28.88 | 46.27 | 6.08 | 11.19 | 25.53 |
| Ours | 13.75 | 21.08 | 45.24 | 5.98 | 13.26 | 29.09 |

As shown in Table 5, we include additional comparisons to Ambient Diffusion (37) across various corruption conditions and datasets. The performance gap between the proposed method and Ambient Diffusion narrows under higher levels of corruption, likely due to the simpler DDPM architecture we employ. In contrast, Ambient Diffusion utilizes the improved DDPM++ architecture (54), which is specifically modified to perform better for high-corruption inpainting-type problems. Nonetheless, the overall performance demonstrates the effectiveness of the proposed method, as we do not focus on empirically optimized architecture details, but instead show the applicability of the new EM idea for training DMs from corrupted observations.

# D  Generative Samples

EMDiffusion is proposed to learn clean distributions from corrupted observations. In Sec. 5, we present detailed posterior sampling results and FID scores of learned DMs.

Our model outperforms baselines by a significant margin in three inverse imaging tasks on two datasets. Though the FID score of our model trained on blurry CelebA is slightly higher than Ambient Diffusion, we argue that FID scores are easily influenced by sharp artifacts introduced by DPS (13), which is adopted in our E-steps. However, the distributions EMDiffusion learned from various types of corrupted observations are obviously better than baselines, as shown in Fig. 7,6,8.

**Future work.** To achieve a better posterior distribution through the proposed EM framework, an accurate and efficient E-step plays a key role. We adopt the Diffusion Posterior Sampling (13) that could potentially introduce artifacts due to its approximation of the underlying data likelihood term. Therefore, a better FID score could be achieved by designing a principled posterior sampling method.

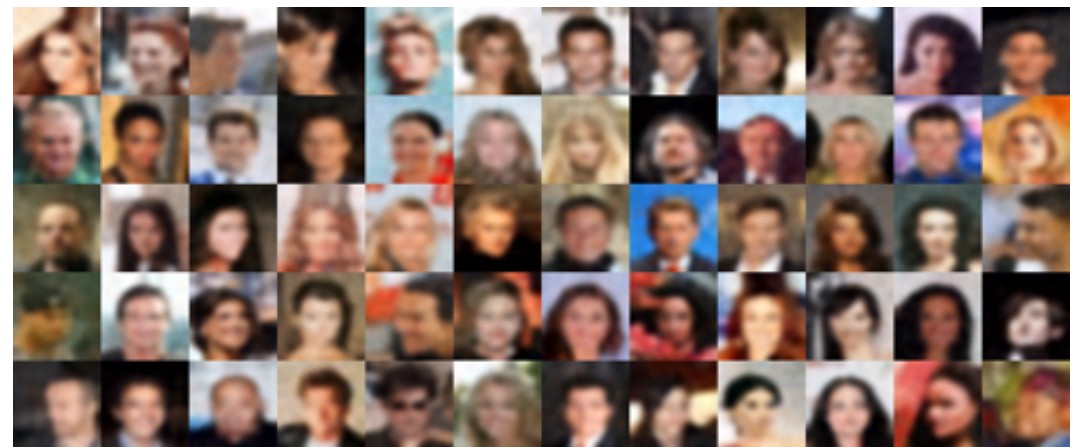

(a) SURE-Score, FID=191.96

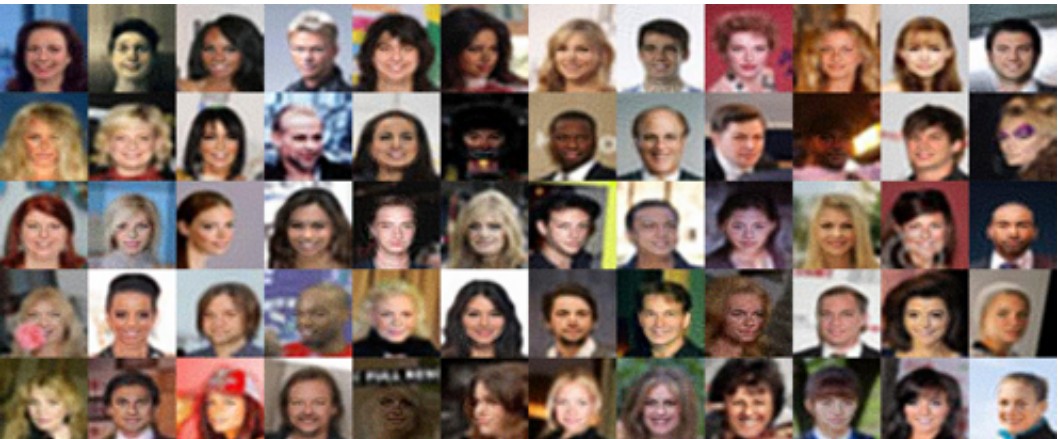

(c) Ours, FID=91.89

Figure 6: Uncurated Samples generated from models trained on blurry CelebA.

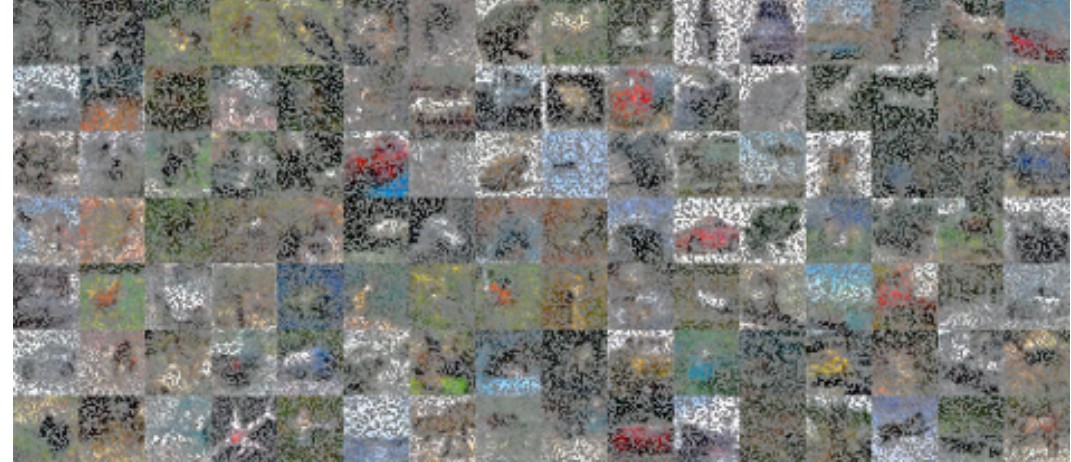

(a) SURE-Score, FID=220.01

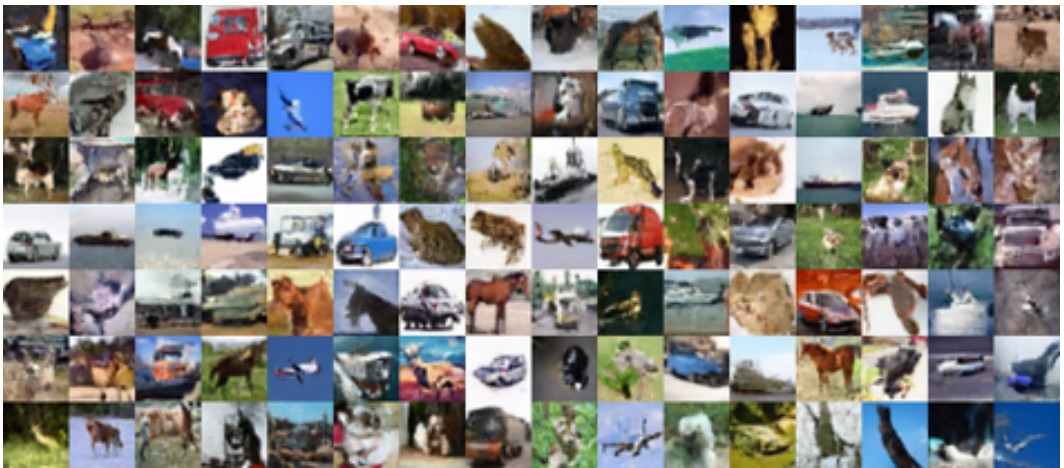

(b) Ambient Diffusion, FID=28.88

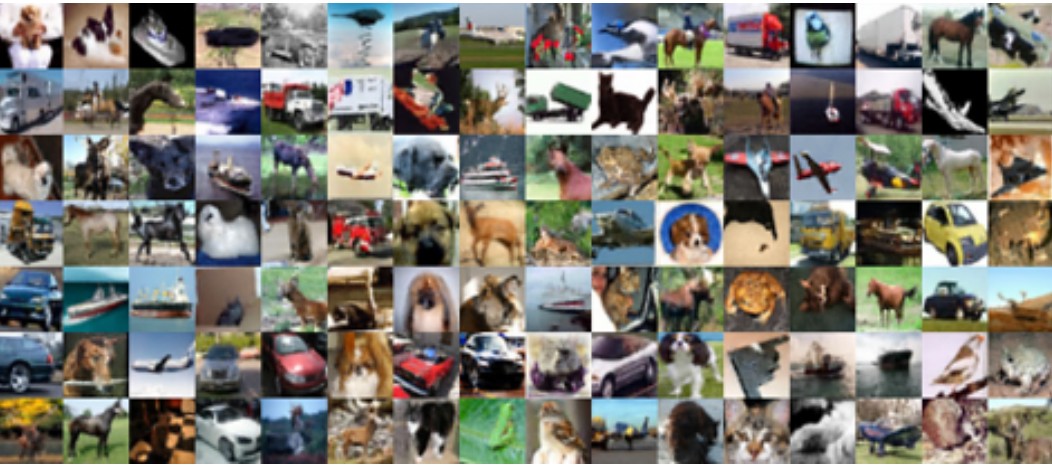

(c) Ours, FID=21.08

Figure 7: Uncurated Samples generated from models trained on random masked CIFAR-10.

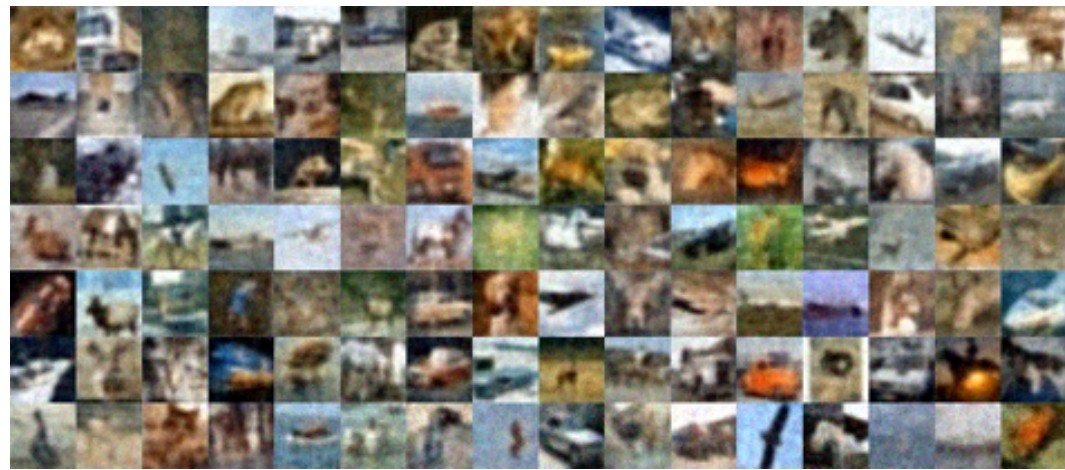

(a) SURE-Score, FID=132.61

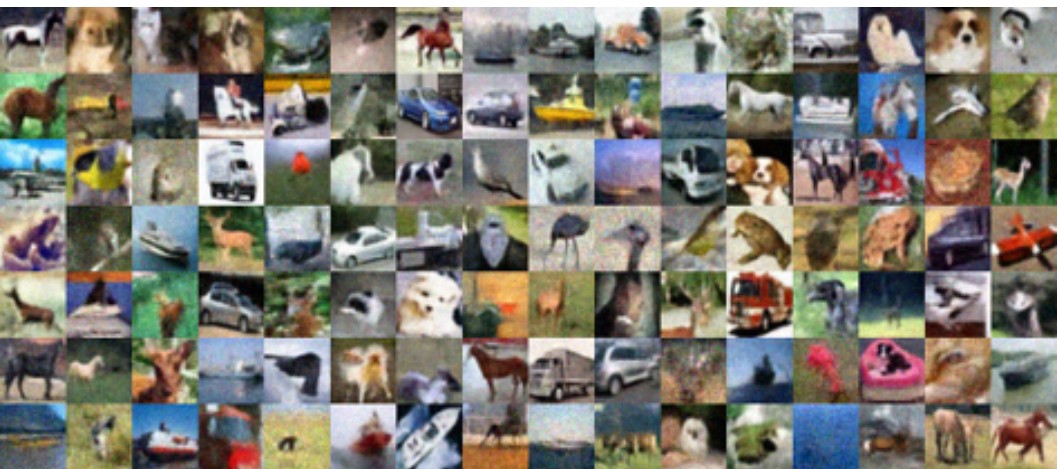

(c) Ours, FID=86.47

Figure 8: Uncurated Samples generated from models trained on noisy CIFAR-10.

