# OpenReview forum: "An Expectation-Maximization Algorithm for Training Clean Diffusion Models from Corrupted Observations"
_NeurIPS.cc/2024/Conference — NeurIPS 2024 poster_

### Official Review · Reviewer_bEKL · 2024-07-11

**Soundness:** 3
**Presentation:** 3
**Contribution:** 3
**Rating:** 7
**Confidence:** 3

**Summary:**

The authors present a new method for training diffusion models from corrupted data.
The presented method is an EM algorithm that in turn (i) uses the current model to sample clean images given the noisy input and then (ii) use these clean samples to train the diffusion model.
The authors show the results for data corrupted by noise, blurring and missing pixels.

**Strengths:**

* This is a conceptual paper that does not focus on empirically optimised architecture details, but instead shows the applicability of a simple elegant new idea. This is a plus.
* I can think of many applications in e.g. in scientific imaging (e.g. microscopy or astronomy) that suffer heavily from noise and degradation and for which clean data training is usually not available or very limited. So, I believe such methods can have a substantial practical impact.
* The paper is well structured, with each building block (Score based diffusion models, posterior sampling, EM algorithm) being carefully introduced.

**Weaknesses:**

* Most importantly, I think the evaluation is not as convincing as it could be. All qualitative results seem to have artefacts to a degree. This also seems to be true for the baselines. E.g. the ambient diffusion results in Figure 3 suffer from some form of pixel artefacts. This is not the case when looking at the ambient diffusion paper (e.g. Figure 7), where not such artefacts are visible. Granted, this is a different dataset. However, I wonder if there is a problem with the training.

* I think the fact that the method requires some (albeit little) clean data as a first training step is a weakness of the method.
* My understanding is that the current system assumes Gaussian noise as part of the forward model (line 149). This can be limiting.
* There are some typos and language problems.

**Questions:**

At the moment the degradation model is known and seems to have to involve Gaussian noise (line 149).
How does this work for the blurring operation, which seems to be deterministic? Or is the amount of added Gaussian noise just not visible.
How does it work for the in-painting task, which also does not seem to involve Gaussian noise?


Typos and language:

Line 103-104 "we defines"
Table 4: "The results show that EM-Score is insensitivity to initializations"

**Limitations:**

Do the authors think that it could be possible to include other degradation models, such as a more realistic combination of Gaussian noise Poisson shot noise?
Would it potentially be possible to learn the parameters of a degradation model. e.g. the blur kernel of the parameters of the noise distribution?
A learnable degradation model would drastically increase the applicability in real life imaging applications.

---

> ### Author Rebuttal · Authors · 2024-08-07
>
> &nbsp;
>
> We thank the reviewer for the constructive suggestions. We are happy that the reviewer recognizes our idea as "simple" and "elegant", our method "can have a substantial practical impact", and our paper "is well structured". We have provided explanations and clarifications regarding all the concerns, as shown below:
>
> &nbsp;
>
> **1. Qualitative results with artifacts**
>
> &emsp; Thanks for your careful review! In Figure 3, we used the officially released Ambient Diffusion checkpoint for image posterior sampling rather than training the SOTA models ourselves. We double-checked our code and found no issues.
>
> &emsp;  However, training clean diffusion models from corrupted images is extremely challenging, and the missing information can cause qualitative results on some datasets to appear artefactual. The good performance in Figure 7 of the Ambient Diffusion paper leverages AFHQ, whereas our results report on CIFAR10. CIFAR10 images contain more high-frequency features than AFHQ images, making inverse problems more challenging and leading to more reconstruction artifacts. The original Ambient Diffusion paper also noted similar issues, stating that learning clean diffusion models on CIFAR10 is harder compared to CelebA-HQ and AFHQ (Sec. 5.1 in their paper) :
> > “for CelebA-HQ and AFHQ, we manage to maintain a decent FID score even with 90% of the pixels deleted. For CIFAR-10, the performance degrades faster, potentially because of the lower resolution of the training images”.
>
> &nbsp;
>
> **2. Dependence on clean data for initialization**
>
> &emsp;  Our method relies on a small amount of clean data as starting point, which we acknowledge as a limitation, as discussed in the conclusion. However, we have minimized this requirement to as few as 10 clean images (Fig. 5(a)). Additionally, we have explored alternative initialization strategies, including using out-of-distribution (OOD) clean images or classical pre-processing methods to initialize EM iterations (see Fig.5 and Tbl. 4 in the main paper). Our preliminary findings suggest that our EM approach can be initialized using low-frequency image statistics (e.g., smoothness) learned from OOD or pre-processed images. Investigating the elimination of the clean data requirement is an important research direction that we leave for future work. We will discuss this further in the revision.
>
> &nbsp;
>
> **3. Gaussian noise in forward models**
>
> &emsp;  All examples in the paper involve Gaussian noise in their degradation models, including inpainting and deblurring, as practical imaging problems always include measurement noise. However, the noise added to masked and blurry images is minimal (std=0.0001 and 0.02, respectively) and may not be visible. We also support different noise models, as discussed below.
>
> &nbsp;
>
> **4. Extending to other types of measurement noise**
>
> &emsp;  Our method can be extended to more realistic noise models. As discussed in the original DPS paper [1], it can handle various noise statistics, such as Gaussian and Poisson. Our EM framework has flexibility in replacing our E-step with DPS under new noise assumptions or even more advanced posterior sampling algorithms such as plug-and-play Monte Carlo (PMC)[2].
>
> &nbsp;
>
> **5. Learning parameters of degradation models**
>
> &emsp; Thank you for suggesting this important topic! Learning degradation models complements our method, which focuses on understanding the relationship between observations and underlying images rather than just image priors. This challenge can also be addressed using the EM framework, where the M-step iteratively estimates the degradation model’s parameters by incorporating the gradient of the data fidelity term with respect to these parameters. We will discuss it in the paper and continue to explore this exciting direction in the future.
>
> &nbsp;
>
> We thank the reviewer once again for the insightful suggestions. We hope our explanations have clarified all the concerns. We are happy to address any further questions or discussions the reviewer may have. Thank you!
>
> &nbsp;
>
> **Reference**:
>
> [1] Hyungjin Chung, Jeongsol Kim, Michael T. Mccann, Marc L. Klasky, Jong Chul Ye. Diffusion Posterior Sampling for General Noisy Inverse Problems, ICLR 2023.
>
> [2] Yu Sun, Zihui Wu, Yifan Chen, Berthy Feng, and Katherine L. Bouman. Provable Probabilistic Imaging using Score-based Generative Priors, arXiv 2023.

---

> ### Comment · Reviewer_bEKL · 2024-08-12
> **Thank you for your reply.**
>
> I don't have any additional questions.

---

> > ### Author Response · Authors · 2024-08-13
> >
> > Thank you so much for your response! We appreciate your time and consideration.

---

### Official Review · Reviewer_Kirh · 2024-07-12

**Soundness:** 4
**Presentation:** 3
**Contribution:** 3
**Rating:** 7
**Confidence:** 3

**Summary:**

This paper we proposes an expectation-maximization (EM) approach to train diffusion models from corrupted observations. The extensive experiments show the effectiveness of the proposed method.

**Strengths:**

1. The experiments are extensive.
2. The proposed approach addresses a significant challenge in the field where large clean datasets are often unavailable.

**Weaknesses:**

1. The source code is not provided yet, but the authors promised to make it public in the future.

**Questions:**

While you mention that only a small number of clean images are needed to initialize the EM iterations, does the number of clean images affect the final model performance in applications? Is it possible to completely eliminate the dependence on clean data?

**Limitations:**

The authors presented the limitations and proposed the future direction to addressed them.

---

> ### Author Rebuttal · Authors · 2024-08-07
>
> &nbsp;
>
> We thank the reviewer for the insightful suggestions. We are happy that the reviewer recognizes our approach "addresses a significant challenge in the field", and our experiments are "extensive" and "show the effectiveness of the proposed method". We will release all the code and the checkpoint to the community to stimulate future research. Below we address each question of the reviewer.
>
> &nbsp;
>
> **1. Source code**
>
> &emsp; We are happy to provide a private repository of our code to address your concerns. Following NeurIPS author guidelines,
>
> > If you were asked by the reviewers to provide code, please send an anonymized link to the AC in a separate comment, make sure the code itself and all related files and file names are also completely anonymized.
>
> &emsp; we have sent a link to the AC for anonymity verification and will share it with you once approved.
>
> &nbsp;
>
> **2. Impact of number of clean images on model performance**
>
> &emsp;  Preliminary experiments in Section 5.4 and Figure 5(a) show that using 10 to 500 clean images results in similar posterior sampling outcomes after the first EM stage, indicating that the diffusion model successfully learns important image statistics with limited data. Since clean images initialize the diffusion model by providing low-frequency statistics, the model performance will not be significantly affected by their number.
>
> &nbsp;
>
> **3. Completely eliminating the dependence on clean data**
>
> &emsp; This is indeed an important research direction. In this paper, we have explored strategies to eliminate the dependence on clean data, such as initializing EM iterations with classical pre-processing methods, as detailed in Appendix B. Additionally, leveraging a generative foundation model like Stable Diffusion as the initial model could be a potential alternative solution, as suggested by other reviewers (Consistent Diffusion Meets Tweedie, ICML 2024). We will continue to explore this direction in future work.
>
> &nbsp;
>
> We thank the reviewer once again for the insightful suggestions. We greatly appreciate the recognition and are more than willing to address any further questions or discussions the reviewer may have. Thank you!

---

> > ### Comment · Reviewer_Kirh · 2024-08-08
> > **My concern is resolved**
> >
> > Thanks for your reply. My concern is resolved.

---

> > > ### Author Response · Authors · 2024-08-11
> > >
> > > We are very happy that the concerns are addressed!  Thank you once again for your valuable feedback!

---

### Official Review · Reviewer_HYFc · 2024-07-13

**Soundness:** 2
**Presentation:** 3
**Contribution:** 2
**Rating:** 2
**Confidence:** 3

**Summary:**

Authors propose using the EM algorithm to train a diffusion model on corrupted data. To initialize the process, the proposed method requires access to a "limited number of clean samples." The authors claim the method allows convergence to the true data distribution.

**Strengths:**

**Originality.** The idea of using EM to train a diffusion model from corrupted data is specifically original; however, training generative models without access to clean data is a well-explored problem, even before the diffusion model era.

**Quality and clarity.** The paper is easy to follow, and the problem is well-motivated.

**Significance.** I do not find the proposed method in its current format significant, as it assumes access to a "limited dataset of clean images" without quantifying how small this set can be. Additionally, the paper claims convergence to the true data distribution without providing conditions on the forward operator and noise process, which is an unsubstantiated claim given the limited set of inverse problems explored.

**Weaknesses:**

* Claiming convergence to the true data distribution is a grand claim, and simply providing examples from relatively simple inverse problems does not substantiate this claim. Therefore, due to unsubstantiated claims and limited quantitative discussion on the required conditions for the forward operator and the number of clean data samples needed, I find this paper not truly usable in practice.

**Questions:**

* How do authors quantify "minimal clean images"? The authors mention
  that sparse clean data provides a good initialization of the DM's
  manifold. Why is this the case? What does "good" mean precisely in
  this context?

**Limitations:**

* I disagree with the authors that needing access to a minimal number of
  clean images is a minor limitation. AmbientDiffusion and some other prior
  work discussed do not assume access to such a dataset, and hence the
  comparisons might not be entirely fair.

---

> ### Author Rebuttal · Authors · 2024-08-07
>
> &nbsp;
>
> We thank the reviewer for the thoughtful comments. We believe there may have been some misunderstandings regarding our work, which we will clarify. Below we carefully address all the questions from the reviewer.
>
> &nbsp;
>
> **1. Quantifying the number of required clean images**
>
> &emsp;  We totally agree that quantifying the amount of required clean images is important and we have carefully investigated this problem in Section 5.4, Figure 5(a), and Table 4 of the main paper. Our experiments show that diffusion models trained on as few as 10 clean images provide adequate initialization. Moreover, to further reduce the need for clean data, we also explored using out-of-distribution (OOD) or pre-processed images for initialization, which achieved comparable results. We will highlight these findings more clearly in the revision.
>
> &nbsp;
>
> **2. Convergence claims & conditions for the forward operator**
>
> &emsp;  In our paper, convergence refers to reaching a local minimum of the optimization problem, as established by the properties of the classical EM algorithm and supported by previous theoretical studies [1]. We discuss this briefly in Section 3.3, lines 124-126, of the original paper. The forward operator and noise process do not affect this type of convergence but do influence the convergence speed by determining the amount of information in the corrupted observations. We will clarify the sentences and definitions about convergence in the final version.
>
> &nbsp;
>
> **3. Why sparse clean data provide “good” initialization**
>
> &emsp;  This question has been partially addressed in Section 4.2, lines 160-165, of the original paper. Sparse clean data helps the initial diffusion model learn common structures in natural images, such as continuity, smoothness, and object profiles. In this context, “good” means capturing important low-frequency statistics of natural images. From the EM algorithm perspective, “good” initialization means defining a local minimum solution close to the global minimum.
>
> &nbsp;
>
> **4. Limitation of requiring clean images**
>
> &emsp;  Considering the minimal number of clean images required for initialization and the alternative strategies we've validated for our EM framework (as explained in answer (1)), we respectfully disagree that this is a major limitation of our algorithm. Additionally, in many scientific applications, accessing some clean images is reasonable, as the lack of clean images is often due to cost rather than physical restrictions. For instance, in fluorescent microscopy, while high-SNR images are limited due to the need for long exposure times and the risk of phototoxicity, it is still feasible to obtain some high-SNR images to support the training of clean diffusion models from extensive low-SNR microscopy data.
>
> &nbsp;
>
> We thank the reviewer once again for the valuable comments. We hope our explanations have clarified all the questions. We are happy to address any further concerns or discussions the reviewer may have. We kindly hope the reviewer could consider raising the score if satisfied with our responses. Thank you!
>
> &nbsp;
>
> **Reference**:
>
> [1] Dempster, A.P.; Laird, N.M.; Rubin, D.B. Maximum Likelihood from Incomplete Data via the EM Algorithm. Journal of the Royal Statistical Society, 1977

---

> > ### Comment · Reviewer_HYFc · 2024-08-12
> >
> > I thank the authors for their response. I will maintain my score because of one main reason: the claims made in the paper are too general and misleading, e.g., "This iterative process leads 8 the learned diffusion model to gradually converge to the true clean data distribution" (from abstract). In the rebuttal, the authors state that "In our paper, convergence refers to reaching a local minimum of the optimization problem." I do not get that impression from the paper. In addition, the authors, in response to my concern regarding conditions on the inverse problem's forward operator, state that "The forward operator and noise process do not affect this type of convergence but do influence the convergence speed by determining the amount of information in the corrupted observations." I strongly disagree: the forward operator could be an all-zeros matrix. You cannot recover the true data distribution using observations like that.

---

> ### Author Response · Authors · 2024-08-13
>
> &nbsp;
>
> We thank the reviewer for the additional comments. We are pleased that many of your concerns have been addressed. Below, we provide additional clarification regarding the convergence claim and the conditions of the forward operator.
>
> &nbsp;
>
> **1. Convergence Claim**
>
> &emsp; As explained in our rebuttal, the convergence we refer to is the local convergence inherited from the EM framework. We explicitly mentioned this in our original paper (lines 124-126). This claim is further supported by our extensive experimental results across various imaging tasks (see Appendix C). We promise to rephrase our abstract and add further clarifications on this point in the revised manuscript to avoid any confusion.
>
> &nbsp;
>
> **2. Forward Operator**
>
> &emsp; In our previous responses, we misunderstood the reviewer's comments as asking us to discuss the effects of various corruption types (e.g., inpainting, denoising, deblurring) on the method’s effectiveness. We fully agree that there are certain cases where recovering a clean distribution from corrupted data is information-theoretically impossible, such as when the forward operator matrix is entirely composed of zeros. We will include additional discussions on these scenarios in the revised manuscript.
>
> &nbsp;
>
> We hope our explanations address all the concerns of the reviewer. We would greatly appreciate it if the reviewer could reconsider the evaluation of our paper. Thank you!

---

### Official Review · Reviewer_qpvz · 2024-07-15

**Soundness:** 4
**Presentation:** 2
**Contribution:** 3
**Rating:** 6
**Confidence:** 5

**Summary:**

The authors introduce a new framework for training diffusion models from corrupted data. Prior work on this research topic is based on the Ambient Diffusion framework or Stein's Unbiased Risk Estimate (SURE) idea. The authors propose an alternative methodology based on the Expectation-Maximization algorithm. The algorithm is tested on standard datasets and leads to improved performance over the considered baselines.

**Strengths:**

1. The research topic of training diffusion from corrupted data is relevant and timely.
2. The authors propose a fresh idea for this interesting research problem. The proposed idea has several elegant properties: I) it is simple, ii) relatively easy to implement, and iii) the idea is oblivious to the type of corruption.
3. The fact that the proposed algorithm is agnostic to the type of corruption is important: as the authors correctly mention, prior work, such as Ambient Diffusion, is designed (or at least tested) only for inpainting.
4. The authors manage to outperform the prior state-of-the-art, Ambient Diffusion in the same setting (CIFAR-10 with 60% corruption).
5. The authors have a nice way of leveraging clean samples. Prior work assumes that we don't have any access to clean samples. This paper naturally integrates the available clean data by using them to initialize the EM algorithm.

Overall, I believe this submission offers a promising alternative to Ambient Diffusion for learning diffusion models with corrupted data.

**Weaknesses:**

1. The comparison to the baselines is somewhat limited/unfair. As the authors mention, Ambient Diffusion has been designed for inpainting. The authors use Ambient Diffusion for denoising and Deblurring. It is not clear how they do this since the original algorithm cannot be trivially extended to these settings. Even for the case of inpainting, the authors could report results for other levels of corruption as well (apart from $p=0.6$) and for other datasets apart from CIFAR-10. Regarding the comparisons with Ambient Diffusion, the only valid data point right now is for $p=0.6$. Providing a more comprehensive analysis would help the readers understand in what regime the proposed method works the best.
2. Specifically for the case of denoising, the authors might want to provide comparisons with the more recent work Consistent Diffusion Meets Tweedie which extends the Ambient Diffusion idea to the setting of noisy samples.
3. For some scientific applications, there is no available clean data at all. The proposed algorithm needs some clean data to initialize the EM and this could be limiting for such applications.
4. One issue with the proposed method is that it relies on solving inverse problems with diffusion models. This problem is computationally intractable and for that reason, there has been a large body of research papers proposing approximations to the conditional score. The authors rely on such an approximation (DPS) which is known that doesn't truly offer samples from the posterior unless the data distribution is very simple (e.g. Gaussian).
5. One issue with this method seems to be the training speed. Even for converge to a local optimum, EM typically requires many steps. Each step here is a new training of a diffusion model. Prior works on learning diffusion models from corrupted data only required a single training.
6. For some corruption types, it is impossible to ever reconstruct the true distribution from noisy observations. The authors do not discuss how the proposed algorithm would work in such cases.
7. Minor: There is a block that needs to be deleted from the NeurIPS checklist section.

**Questions:**

See the Weaknesses Section above. I am willing to further increase my score if my concerns are properly addressed.

**Limitations:**

The authors have adequately addressed the limitations of their method.

---

> ### Author Rebuttal · Authors · 2024-08-07
>
> &nbsp;
>
> We thank the reviewer for the invaluable feedback. As noted by the reviewer, our idea is "fresh" and "elegant," our algorithm is "agnostic to the type of corruption," and our results "outperform the prior state-of-the-art."  Below we provide more explanations and additional results to address each concern from the reviewer.
>
> &nbsp;
>
> **1. Comparison to baselines**
>
> &emsp;  Ambient Diffusion pioneered in learning clean diffusion models from masked images, achieving current STOA performance. While this insightful idea was originally designed for inpainting task, we feel it is more comprehensive to test it on denoising and deblurring tasks, using its original implementation with a naïve further corruption technique (randomly masking pixels for all tasks). We acknowledge that the original algorithm cannot be trivially extended to different settings. To avoid misinterpretation, we are open to labeling Ambient Diffusion's denoising and deblurring results as "not applicable," as these tasks fall outside its original design scope. In addition, we also provided SURE-Score in our paper as a versatile baseline capable of handling various corruptions.
>
> &emsp; We agree that validation on different levels of corruption and other datasets would strengthen the understanding of our method. As suggested by the reviewer, we conducted additional experiments comparing the EM approach with Ambient Diffusion across different corruption levels (masking ratios of 0.4, 0.6, 0.8). These new results, presented in the table below and the attached PDF, demonstrate our method’s robustness and superior performance across diverse conditions. We will incorporate these findings as well as more experiments on new datasets (e.g., CelebA) in the revision.
>
> | Masking Ratio                     | 0.4   | 0.6   | 0.8   |
> | --------------------------------- | ----- | ----- | ----- |
> | AmbientDiffusion, FID$\downarrow$ | 18.85 | 28.88 | 46.27 |
> | Ours, FID$\downarrow$             | 13.75 | 21.08 | 45.24 |
>
>
> &nbsp;
>
> **2. Recent work "Consistent Diffusion Meets Tweedie"**
>
> &emsp; We thank the reviewer for pointing out this relevant ICML 2024 work, which cleverly finetunes Stable Diffusion (SD) to leverage the pre-trained knowledge in denoising tasks. We will add it in related work. However, as our method focuses on training diffusion models from scratch using corrupted observations, we feel the two works belong to different categories. A fairer comparison with the ICML work would involve applying an EM framework to finetune SD with noisy observations, which presents intriguing future research. We will explore it in future work.
>
> &nbsp;
>
> **3. The limitation of clean data initialization**
>
> &emsp; Our method relies on a small amount of clean data as a starting point, which we do acknowledge as a limitation, as discussed in the conclusion. However, we have minimized the requirement of the clean data to as few as 10 or 50 clean images (Fig. 5(a)), which we believe is a reasonable assumption for many scientific applications. For instance, in fluorescent microscopy imaging, high SNR images are limited due to long exposure times and potential phototoxicity rather than physical restrictions. Therefore, acquiring a small number of high-SNR images to aid training is feasible.
>
> &emsp; On top of that, we have explored alternative initialization strategies, including using out-of-distribution (OOD) clean images or classical pre-processing methods to initialize EM iterations (see Fig. 5 and Tbl. 4 in the paper). Our preliminary findings suggest that our EM approach can be initialized using low-frequency image statistics (e.g., smoothness) learned from OOD or pre-processed images. Investigating ways to eliminate the need for clean data would be an important future research direction. We will discuss more in the revision.
>
> &nbsp;
>
> **4.DPS does not offer true posterior samples**
>
> &emsp; We acknowledge that DPS-based methods may not always accurately sample the true posterior due to their simplified assumptions. Our key contribution is the EM approach itself, which provides flexibility in choosing the posterior sampling method in its E-step. For example,  it can replace DPS with more advanced posterior sampling techniques with stronger theoretical guarantees, such as plug-and-play Monte Carlo (PMC)[1], which ensures non-asymptotic stationarity. We will discuss it in the revision.
>
> &nbsp;
>
> **5. Training efficiency**
>
> &emsp; Although our method involves multiple training steps,  as discussed in Sec. 4.3, we avoid training diffusion models from scratch at each M-step. Instead, we fine-tune the model in most iterations, only training from scratch in the final 1-3 steps to prevent memorizing poor samples from the initial stages. This ensures rapid convergence in each iteration and significantly reduces training time. In practice, our method has a similar training time as Ambient Diffusion (ours is 2 days with 4 NVIDIA A800 GPUs).
>
> &nbsp;
>
> **6. Discussion on corruption types that are impossible to recover**
>
> &emsp;  We agree that for extremely corrupted images (e.g., >99% pixels masked), reconstructing the true distribution becomes challenging or even impossible due to the loss of spatial relationships among pixels. This is a common issue across all baselines. A potential solution could involve incorporating more prior knowledge, such as fine-tuning SD instead of training from scratch, similar to ICML 2024 work. We will discuss it in the revision.
>
> &nbsp;
>
> **7.Redundancy in the checklist section**
>
> &emsp;  Nice catch! We will correct it.
>
> &nbsp;
>
> We thank the reviewer again for the valuable insights and suggestions. We hope our explanations have clarified all the concerns. We are happy to address any further questions or discussions the reviewer may have. Thank you!
>
> &nbsp;
>
> **Reference**:
>
> [1] Yu Sun, Zihui Wu, Yifan Chen, Berthy Feng, and Katherine L. Bouman. Provable Probabilistic Imaging using Score-based Generative Priors, arXiv 2023.

---

> > ### Comment · Reviewer_qpvz · 2024-08-11
> > **Rebuttal acknowledgement**
> >
> > I would like to thank the authors for their time and their efforts during the rebuttal.
> >
> > I believe that the experiments with Ambient Diffusion for tasks beyond inpainting should be completely removed. Extending this method to corruptions beyond random inpainting should be done in a way such that Theorem 4.1. of the paper is satisfied. The current mechanism for further corruption is not valid and hence the results are not valid. I would like to ask the authors to remove the results for such tasks to avoid confusing the readers.
> >
> > I would like to thank the authors for providing additional comparisons to Ambient Diffusion. I would urge the authors to provide more points and more datasets. This is crucial: different proposed methods should be evaluated in the same setting so that future work can be compared on a common benchmark. From the current results, it looks like the gap between the proposed method and Ambient Diffusion is closing for higher corruption. Results for higher corruption should be provided as well. In my opinion, it is absolutely fine if the proposed method even underperforms a baseline in some settings. The contribution is significant already.
> >
> > Regarding the ICML work, it provides a general algorithm for training diffusion models from noisy observations than can be applied to training from scratch as well. In any case, this is concurrent work and I agree with the authors that direct comparisons are not needed.
> >
> > Regarding discussion for other corruption types: this is not what I meant. I was referring to cases where it is information-theoretically impossible to learn the underlying distribution from measurements, e.g. you can't learn the distribution of human faces from a set of training images where the eyes are always missing.
> >
> > Overall, I believe this paper has some really nice aspects and hence I will keep my positive score of 6.

---

> > > ### Author Response · Authors · 2024-08-11
> > >
> > > We sincerely thank the reviewer for the additional comments. We will incorporate all your invaluable suggestions in our revision. Specifically, we will:
> > >
> > > 1） Remove AmbientDiffusion’s results on deblurring and denoising.
> > >
> > > 2） Include experiments on more corruption conditions and datasets.
> > >
> > > 3） Add discussion on the cases where learning clean diffusion from corrupted data is theoretically impossible.
> > >
> > > Thank you once again for your valuable feedback!

---

### Author Rebuttal · Authors · 2024-08-07

&nbsp;

We thank all the reviewers for their professional and constructive feedback. We are encouraged by their recognition of our paper's technical importance and novelty (qpvz, Kirh, bEKL), broad applicability (qpvz, bEKL), impressive performance in extensive experiments (qpvz, Kirh), and high-quality writing (HYFc, bEKL).   We have summarized some of the major concerns raised by the reviewers below. Point-to-point responses are also included as a reply to each reviewer. Additionally, we have added a PDF file for further experimental results.

&nbsp;

**1: Requirement for clean data to initialize the EM training**

&emsp; The introduction of clean images aids the diffusion model in converging to a local minimum close to the true image distribution. We clarify that as few as 10 images may suffice, and alternative initialization strategies also exist, as discussed in Section 5.4 and Appendix B of our original submission. We acknowledge that other strategies suggested by reviewers, such as initializing with generative foundation models (e.g., Stable Diffusion[1]), are promising and will discuss them in the revision. We also discuss scientific imaging applications (e.g., fluorescent microscopy) to explain that access to some clean images is reasonable, as their scarcity is often due to cost rather than physical restrictions.

&nbsp;

**2: More comparisons to baseline methods like Ambient Diffusion**

&emsp;  While our original submission included extensive experiments comparing our method with baselines, as suggested by the reviewers, we added additional experiments with different settings (e.g., various corruption levels) and further verification of our implementation to strength our conclusions. We have included these new results and the corresponding conclusions in the PDF as well as the point-by-point responses.

&nbsp;

**3: The algorithm’s extension to different corruption types, noise models, and unknown degradations**

&emsp;  The EM algorithm is a versatile framework that can be extended to various imaging inverse problems and can also learn unknown forward models or noise statistics. We will include more discussions on these aspects in our revision.

&nbsp;

Please let us know if these clarifications and additional results address your concerns. We are happy to discuss any remaining points during the discussion phase.

&nbsp;
&nbsp;

**Reference**:

[1] Robin Rombach, Andreas Blattmann, Dominik Lorenz, Patrick Esser, Björn Ommer. High-Resolution Image Synthesis with Latent Diffusion Models, CVPR 2022.

---

### Decision · Program_Chairs · 2024-09-25

**Decision:**

Accept (poster)

**Comment:**

This paper received mixed ratings: Weak Accept, Strong Reject, Accept, Accept. The authors propose an expectation-maximization (EM) approach to train diffusion models from corrupted observations. AC agrees with some reviewers that the proposed is a simple yet elegant idea to alternate between reconstructing clean images from corrupted data using a known diffusion model (E-step) and refining diffusion model weights based on these reconstructions (M-step). Additional strengths highlighted by reviewers include the proposed method is agnostic to the type of corruption, and potentially applicable to many applications, e.g., in scientific imaging. AC agrees with Reviewer bEKL that "This is a conceptual paper that does not focus on empirically optimised architecture details, but instead shows the applicability of a simple elegant new idea." Reviewer HYFc raises a major concern that "claiming convergence to the true data distribution is a grand claim". The authors provided an acceptable response, and agreed to further clarify the sentences and definitions about convergence in the final version. AC recommends accepting the paper.